# The Cystic Fibrosis Transmembrane Conductance Regulator Gene (CFTR) Is under Post-Transcriptional Control of microRNAs: Analysis of the Effects of agomiRNAs Mimicking miR-145-5p, miR-101-3p, and miR-335-5p

**DOI:** 10.3390/ncrna9020029

**Published:** 2023-04-18

**Authors:** Chiara Papi, Jessica Gasparello, Matteo Zurlo, Lucia Carmela Cosenza, Roberto Gambari, Alessia Finotti

**Affiliations:** 1Department of Life Sciences and Biotechnology, Section of Biochemistry and Molecular Biology, University of Ferrara, 44121 Ferrara, Italy; 2Research Center on Innovative Therapies for Cystic Fibrosis, University of Ferrara, 44121 Ferrara, Italy

**Keywords:** microRNAs, miR-145-5p, CFTR, miRNA therapy, agomiRNAs, pre-miRNAs

## Abstract

(1) Background: MicroRNAs are involved in the expression of the gene encoding the chloride channel CFTR (Cystic Fibrosis Transmembrane Conductance Regulator); the objective of this short report is to study the effects of the treatment of bronchial epithelial Calu-3 cells with molecules mimicking the activity of pre-miR-145-5p, pre-miR-335-5p, and pre-miR-101-3p, and to discuss possible translational applications of these molecules in pre-clinical studies focusing on the development of protocols of possible interest in therapy; (2) Methods: *CFTR* mRNA was quantified by Reverse Transcription quantitative Polymerase Chain Reaction (RT-qPCR). The production of the CFTR protein was assessed by Western blotting; (3) Results: The treatment of Calu-3 cells with agomiR-145-5p caused the highest inhibition of *CFTR* mRNA accumulation and CFTR production; (4) Conclusions: The treatment of target cells with the agomiR pre-miR-145-5p should be considered when *CFTR* gene expression should be inhibited in pathological conditions, such as polycystic kidney disease (PKD), some types of cancer, cholera, and SARS-CoV-2 infection.

## 1. Introduction

MicroRNAs (miRNAs) are a class of short (less than 30 nucleotides in length) non-coding RNAs with a very important role in the regulation of gene expression at the post-transcriptional level [1,2,3] through sequence-specific interactions with the 3′UTR of target mRNAs. This causes translational repression or mRNA degradation [2]. Several miRNA/mRNA networks are responsible for the control of important biological functions, such as cell growth, apoptosis, and differentiation [3], and their alterations are associated with the onset and/or progression of several human pathologies [4,5,6]. In agreement, the alteration of miRNA biological functions (either by miRNA inhibition or miRNA mimicking) is considered an interesting and innovative strategy with potential therapeutic implications [7,8,9,10]. This is not merely a theoretical possibility since efforts aimed at translating laboratory investigations to clinical practice are in progress, as demonstrated by the several ongoing clinical trials such as NCT01646489, NCT01200420, NCT02826525, and NCT02580552, and by the increasing number of reports and reviews related to this issue [11,12,13].

Several studies have been reported demonstrating that microRNAs are involved in the expression of the gene encoding the chloride channel CFTR (Cystic Fibrosis Transmembrane Conductance Regulator) [14,15,16,17,18,19].

Our group has reported that miR-145-5p [20,21,22], miR-101-3p [23], and miR-335-5p [24] play an important role in regulating the expression of the *CFTR* gene. In particular, miR-145-5p has been demonstrated to regulate *CFTR* by several research groups as well, as demonstrated by Oglesby et al. [19], Lutful Kabir et al. [25], and Dutta et al. [26]. A possible therapeutic application of these studies is that targeting these miRNAs (in our case miR-145-5p, miR-101-3p, and miR-335-5p) with antagomiRNAs could be considered an experimental strategy to upregulate *CFTR* expression. Interestingly, the activity of different miRNAs on post-transcriptional *CFTR* regulation might be synergistic, as found by Megiorni et al. [15] for miR-101 and miR-494 and by Papi et al. [27] for miR-145-5p and miR-101-3p. All these studies were in strong agreement in proposing anti-miRNA molecules targeting *CFTR* regulatory miRNAs as molecular agents that are able to upregulate *CFTR*. This was further confirmed using molecular blockers masking the miRNA-binding sites present in the *CFTR* 3′UTR [28,29]. A list of the most relevant regulatory miRNAs interacting with the *CFTR* 3′UTR and a scheme outlining the antisense and miRNA mimicking approach to control *CFTR* gene expression are presented in Figure 1.

AgomiR molecules mimicking the activity of the *CFTR* regulatory miRNAs are, on the contrary, expected to downregulate *CFTR* [9,19].

The objective of this short report is to comparatively characterize the possible effects of treatment of bronchial epithelial Calu-3 cells [20,27] and intestinal Caco-2 cells [30] with agomiRNA molecules mimicking pre-miR-145-5p, pre-miR-335-5p, and pre-miR-101-3p, and to discuss possible translational applications of these molecules in pre-clinical studies focusing on the development of protocols of possible interest in therapy.

## 2. Results

### 2.1. Effects of Treatment of Calu-3 Cells with Pre-miRNAs Regulating CFTR Expression

The results shown in Figure 2 demonstrate that CFTR is heavily modulated (downregulated) by some pre-miRNAs mimicking the bioactivity of miRNAs targeting the 3′UTR of the *CFTR* mRNA. The activity of three pre-miRNAs was compared, i.e., pre-miRNA-145-5p, pre-miRNA-101-3p, and pre-miRNA-335-5p. The treatment with high pre-miR concentrations was performed in order to compare the activities of miRNA-145-5p, pre-miRNA-101-3p, and pre-miRNA-335-5p, used at the same concentration, and considering that preliminary results indicated very low activity of pre-miR-101-3p; therefore, it required comparison at a high concentration (i.e., 300 nM). After 72 h of treatment of Calu-3 cells with 100 and 300 nM pre-miRNAs, the cells were harvested and the produced proteins were analyzed by Western blotting using the CFTR antibody against the NBD2 domain of CFTR, clone 596 (University of North Caroline, Cystic Fibrosis Center, Chapel Hill, NC, USA). The antibody specific for the housekeeping of Na+/K+-ATPase (SC-514614, Santa Cruz Biotechnology, Dallas, TX, USA) was employed for normalization. The obtained results show that the treatment with pre-miR-101-3p was ineffective, while the treatment with pre-miR-145-5p and pre-miR-335-5p strongly reduced CFTR production (see the quantitative analyses shown in Figure 2B).

The treatment of Calu-3 cells with miR-145-5p was more effective in inhibiting CFTR production than the treatment with pre-miR-335-5p. Accordingly, we focused our experiments on the use of pre-miR-145-5p. The effects of pre-miR-145-5p were detectable even when the agomiR was used at a concentration of 25 nM (Figure 3).

### 2.2. The Effects of Treatment of Calu-3 Cells with Pre-miR-145-5p Do Not Require Multiple Additions of the agomiR

We have compared two protocols. In the first protocol (protocol #1, Figure 4A–C), Calu-3 cells were treated with 300 nM pre-miR-145-5p (final concentration), and administered at the time of seeding (time 0, see Figure 4A). Further aliquots of pre-miR-145-5p were not added. In the second protocol (protocol #2, Figure 4D–F), three aliquots of pre-miR-145-5p (100 nM final concentrations) were added at the 0 time point and after 24 and 48 h (see Figure 4D). The two protocols were considered taking into account the possibility that the pre-miRNA molecules were unstable, or prone to intracellular compartmentalization associated with loss of bioactivity. In addition, long-lasting activity is required for possible therapeutic applications of biomolecules.

In both protocols, after 72 h of cell culturing, Calu-3 cells were isolated, RNA and proteins were purified, and RT-qPCR and Western blotting analyses were performed for the quantification of *CFTR* mRNA (Figure 4B,E) and CFTR protein (Figure 4C,F).

The obtained results demonstrate that the addition of pre-miR-145-5p at the beginning of Calu-3 cell culture (time 0) is sufficient to obtain the maximum level of inhibition of CFTR expression, analyzed by RT-qPCR (Figure 4B,E) and Western blotting (Figure 4C,F). These results strongly suggest that the agomiR is stable under these experimental conditions.

### 2.3. Pre-miR-145-5p Inhibits CFTR Expression in the Intestinal Cell Line Caco-2

The CFTR function is important not only in the lungs, but also in other tissues, such as the intestine, pancreas, and kidney. In this respect, potential therapeutic strategies to combat defective CFTR function in these organs are being considered [31]. In addition, CFTR should be downregulated in case its overexpression is associated with pathological conditions [32]. For this reason, we have extended the study on the effects of pre-miR-145-5p on the intestinal Caco-2, that are an important cellular system for the screening and validation of CFTR inhibitors [30,33].

The data are shown in Figure 5, which fully support the conclusions presented using the bronchial Calu-3 cells, i.e., that pre-miR-154-5p is a potent (and flexible) inhibitor of the expression of CFTR. 

## 3. Discussion

Non-viral microRNA gene therapy is a molecular intervention based on the treatment of target cells (a) with antagomiRNA molecules able to interfere with the biological activity of microRNAs or (b) with agomiRNA molecules able to mimic the miRNA-associated effects. In the case of treatment with antagomiRNAs, the expected effect is an upregulation of those mRNAs whose expression is controlled by the target miRNAs. Conversely, the use of agomiRNA molecules is expected to cause the downregulation of target mRNAs (see Figure 1B).

There is a general agreement that the Cystic Fibrosis Transmembrane Regulator (CFTR) Gene is under the control of microRNAs [14,15,16,17,18,19,20]. In this context, the major focus of previously published studies [20,21,22,23,24,27] was on hyper-expressing CFTR with the use of antagomiRNA molecules targeting miRNAs downregulating CFTR (for instance, miR-145-5p). This might be of interest for the therapy of Cystic Fibrosis. In those reports, the issue and the translational potential of inhibiting CFTR were not considered. On the contrary, one of the main results of this short report is that pre-miR-145-5p is effective in inhibiting CFTR expression with an efficiency higher than pre-miR-335-5p and pre-miR-101-3p (Figure 2). The addition of pre-miR-145-5p at the beginning of the cell culture period is sufficient to obtain the expected biological effect (i.e., CFTR inhibition) (Figure 3). These results allow proposing the employment of pre-miR-145-5p in pre-clinical studies focusing on pathologies in which CFTR inhibition should be considered a clinically relevant target.

The major limitation of our study is that only one delivery system was employed. Therefore, our studies need to be extended to other delivery systems, as those recently reported are considered very useful for biomedical applications [34]. Moreover, a second limit is that co-treatment using co-administration of different pre-miRNAs has not been considered. Hence, this study should be considered as a proof on concept suggesting that future studies are needed to determine whether co-treatment might lead to improving the inhibitory effects on CFTR.

We suggest that the translational applications of our results to “miRNA therapeutics” should be considered a strength of our study; in this context, the use of pre-miR-145-5p deserves further pre-clinical studies that will be performed using appropriate pathological model systems.

The first consideration is that CFTR inhibition might be a useful approach for human pathologies, such as polycystic kidney disease (PKD) [35]. The involvement of CFTR in PKD has been described in several studies [35,36,37,38]. For instance, Davidow et al. found that CFTR mediates transepithelial fluid secretion by human autosomal dominant polycystic kidney disease epithelium in vitro [36]. In agreement, small-molecule CFTR inhibitors slow cyst growth in polycystic kidney disease, as reported by Yang et al. [33]. Examples of CFTR inhibitors that are potentially useful to treat PKD are pinostrobin [38], chalone derivatives [39], and steviol [40].

Even though *CFTR* has been proposed as a tumor suppressor gene [41,42], its upregulation has been described in some types of cancers [43,44,45]. For instance, Liu et al. found that high *CFTR* expression occurs in childhood B-cell acute lymphoblastic leukemia and is a potential therapeutic target [43]. Zhao et al. described that CFTR promotes malignant glioma development via upregulation of the Akt/Bcl2-mediated anti-apoptosis pathway [44]. Finally, Xu et al. [45] reported that high expression of *CFTR* is associated with tumor aggressiveness of ovarian cancer. Accordingly, knockdown and/or inhibition of CFTR suppresses the proliferation of tumor cells in vitro and in vivo [45]. Interestingly, the CFTR inhibitor miR-145-5p is a recognized tumor suppressor miRNA [46,47,48]. These published studies, together with our results, should stimulate research efforts aimed to verify, in CFTR-overexpressing tumor cell lines, the interplay between pre-miR-145-5p treatment, CFTR downregulation, and effects on in vitro tumor phenotype.

A final application of miRNA-mediated downregulation of CFTR is in the field of infectious diseases. The first example is related to the role of CFTR functions in exacerbating secretory diarrhea, which remains an important global health problem and is strictly associated with the activity of the main virulence factor produced by *Vibrio cholerae*, the cholera toxin (CT) [49]. It is accepted that overstimulation of CFTR-mediated Cl- secretion plays an important role in the pathogenesis of secretory diarrheas [32,33,50]. Therefore, biomolecules acting as CFTR inhibitors can be proposed as a potential antisecretory therapy for diarrheas caused by CT [32,33,51,52]. The usefulness and antidiarrheal efficacy of CFTR inhibitors have been validated on human intestinal epithelial cells and in mouse models of cholera [49]. Interestingly, pre-miR-145-5p is a potent inhibitor of CFTR using the Caco-2 intestinal cell line (Figure 5); a model system to validate CFTR inhibitors for possible treatment of secretory diarrhea associated with cholera [33,39].

The second example of possible applications of CFTR downregulation in infectious diseases is related to the role of CFTR in the life cycle of SARS-CoV-2, responsible for the COVID-19 pandemic [53]. A possible relationship between CFTR and COVID-19 has been suggested by the surprising observation that CF patients (who produce low or altered levels of functionally deficient CFTR) are significantly protected against infection by SARS-CoV-2 [54], supporting the hypothesis of a possible role of CFTR inhibitors (such as pre-miR-145-5p) on the spread of COVID-19. Accordingly, Lotti et al. [55] provided evidence that CFTR expression/function was involved in the regulation of SARS-CoV-2 replication. Moreover, Bezzerri et al. were able to demonstrate that ACE2 expression and localization are regulated by CFTR, supporting the concept that molecules mimicking the biological activity of miRNA downregulating CFTR (such as miR-145-5p) should be considered as potential anti-SARS-CoV-2 agents [56]. This was formally demonstrated in SARS-CoV-2 infected Calu-3 cells [56], confirming that miR-145-5p-mediated downregulation of CFTR might be associated with inhibition of the SARS-CoV-2 life cycle. Interestingly, using differential expression analysis of microarray data, hsa-miR-145-5p was identified among the top differentially expressed transcripts in severe versus asymptomatic/mild cases [57]. The microarray data confirmed the miR-145-5p downregulation in severe cases of COVID-19, while miR-145-5p was found upregulated in the mild cases.

## 4. Materials and Methods

### 4.1. Cell Lines and Culture Conditions

The human bronchial epithelial Calu-3 cell line was obtained from Prof. Anna Ta-manini (Laboratory of Molecular Pathology, Department of Pathology and Diagnostics, University Hospital of Verona, Verona, Italy) [24,27,58]; the intestinal epithelial Caco-2 cells were obtained from Prof. Giuseppe Valacchi (Department of Environmental Sciences and Prevention, University of Ferrara, Ferrara, Italy) [30,59]. Calu-3 and Caco-2 cells were cultured in a humidified atmosphere of 5% CO2/air in DMEM/F12 and in a DMEM medium, respectively (Gibco, Thermo Fisher Scientific, Waltham, MA, USA), supplemented with 10% fetal bovine serum (Biowest, Nauillè, France), 100 units/mL penicillin, 100 µg/mL streptomycin (Lonza Bioscience, Verviers, Belgium), and 1% NEEA (100X) (non-essential amino acids solution; Gibco, Thermo Fisher Scientific, Waltham, MA, USA).

### 4.2. Cell Treatments with agomiRNAs

Calu-3 and Caco-2 cell lines were seeded in a 12-well plate with a concentration of 250,000 cells/mL and 50,000 cells/mL, respectively. The cells were treated with pre-miR^TM^ miRNA precursor (Ambion, Thermo Fisher Scientific, Waltham, MA, USA), indicated in Table 1, using Lipofectamine RNAiMAX transfectant reagent (Invitrogen, Thermo Fisher Scientific, Waltham, MA, USA) according to the manufacturer’s instruction. We have performed two different transfection procedures using the Calu-3 cell line.

For protocol #1 cells were treated with pre-miR-145 with a final concentration of 300 nM, whereas in protocol #2, Calu-3 cells were transfected with pre-miR-145 molecules with a concentration of 100 nM at the time of seeding, and this transfection procedure was repeated after 24 and 48 h. In all treatments, after 72 h, the cells were collected, washed with sterile PBS, and the total cell extracts and RNA were prepared for further analysis.

### 4.3. RNA Extraction

Cultured cells were trypsinized and collected by centrifugation at 1500 rpm for 10 min at 4 °C, washed with PBS, and lysed with Tri-Reagent (Sigma Aldrich, St. Louis, MO, USA), according to the manufacturer’s instructions. The isolated RNA was washed once with cold 75% ethanol, dried, and dissolved in nuclease-free pure water before use [20,23,24]. 

### 4.4. Analysis of CFTR Expression: RT-qPCR

The gene expression analysis was performed by RT-qPCR using 500 ng of total RNA, extracted and reverse transcribed using the Taq-Man Reverse Transcription PCR Kit, and random hexamers (Applied Biosystems, Thermo Fisher Scientific, Waltham, MA, USA) as RT reaction primers. Quantitative real-time PCR (RT-qPCR) assays were carried out using gene-specific double fluorescently labeled probes. The primers and probes used to assay *CFTR* (Assay ID: Hs00357011_m1) gene expression were purchased from Applied Biosystems. The relative expression was calculated using the comparative cycle threshold method and, as reference genes, the human *RPL13A* (Assay ID: Hs03043885_g1).

### 4.5. Analysis of CFTR Expression: Western Blotting

The CFTR expression was measured using Western blotting analyses. Cell pellets were lysed in RIPA buffer (Thermo Fisher Scientific, Waltham, MA, USA) and sonicated for 30 sec three times on ice at 50% amplitude using the Vibra-Cell VC130 Ultrasonic Processor (Sonics, Newtown, CT, USA). The lysates were cleared by centrifugation at 14,000× *g* for 30 min at 4 °C. The protein concentration was determined using the BCA Protein Assay Kit (Pierce, Thermo Fisher Scientific, Waltham, MA, USA) according to the manufacturer’s protocol. For the CFTR analysis, 40 μg of total protein extracts were heated in Blue Loading Buffer by adding 50 nM dithiothreitol (DTT) (Cell Signaling Technology, Danvers, MA, USA) at 37 °C for 10 min and loaded onto a 7% SDS-polyacrylamide gel. The gel proteins were transferred to a nitrocellulose membrane (Thermo Fischer Scientific, Waltham, MA, USA) by using Trans-Blot Turbo (Bio-Rad Laboratories, Hercules, CA, USA) and processed for Western blotting by using a mouse monoclonal antibody, clone 596, against the NBD2 domain of CFTR (University of North Carolina, Cystic Fibrosis Center, Chapel Hill, NC, USA) at a dilution of 1:2500 by overnight incubation at 4 °C. After the washes, the membranes were incubated with horseradish peroxidase-coupled anti-mouse immunoglobulin (R&D System, Minneapolis, MN, USA) at room temperature for 1 h, and after washes, the signal was developed by enhanced chemiluminescence (LumiGlo Reagent and Peroxide, Cell Signaling, Danvers, MA, USA). After membrane stripping, a Na+/K+-ATPase monoclonal antibody (sc-514614, Santa Cruz Biotechnology, Dallas, TX, USA) or vinculin monoclonal antibody (sc-73614, Santa Cruz Biotechnology, Dallas, TX, USA) was used to investigate the equal loading of samples [27].

### 4.6. Statistical Analysis

The results are expressed as mean ± standard error of the mean (SEM). Comparisons between groups were made by using one-way ANOVA (* *p* < 0.05, ** *p* < 0.01, *** *p* < 0.001, and **** *p* < 0.0001).

## Figures and Tables

**Figure 1 ncrna-09-00029-f001:**
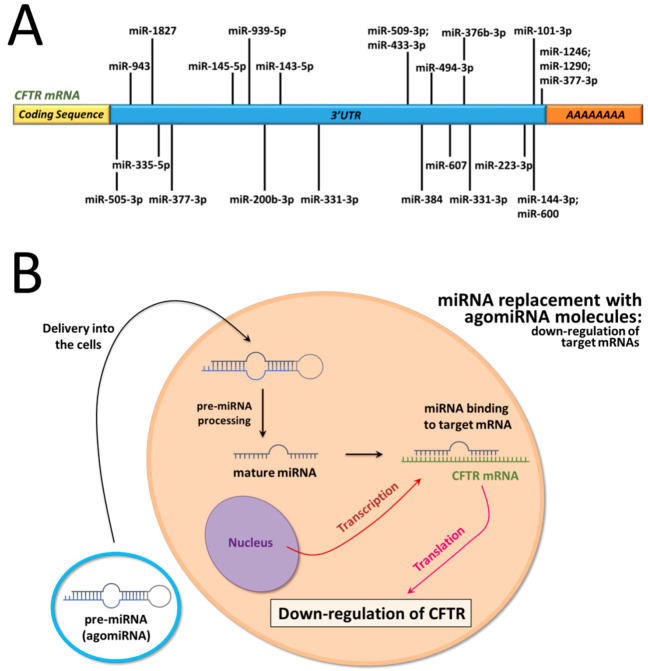
(**A**). Location of miRNA binding sites within the 3′UTR of the *CFTR* mRNA. (**B**). Scheme depicting the miRNA replacement approach using transfected agomiRNAs (see panel **A**) to downregulate *CFTR* gene expression. Adapted with permission from Papi et al. [27]; (copyright can be found at https://www.mdpi.com/1422-0067/23/16/9348 (2022, by Papi et al.) [27].

**Figure 2 ncrna-09-00029-f002:**
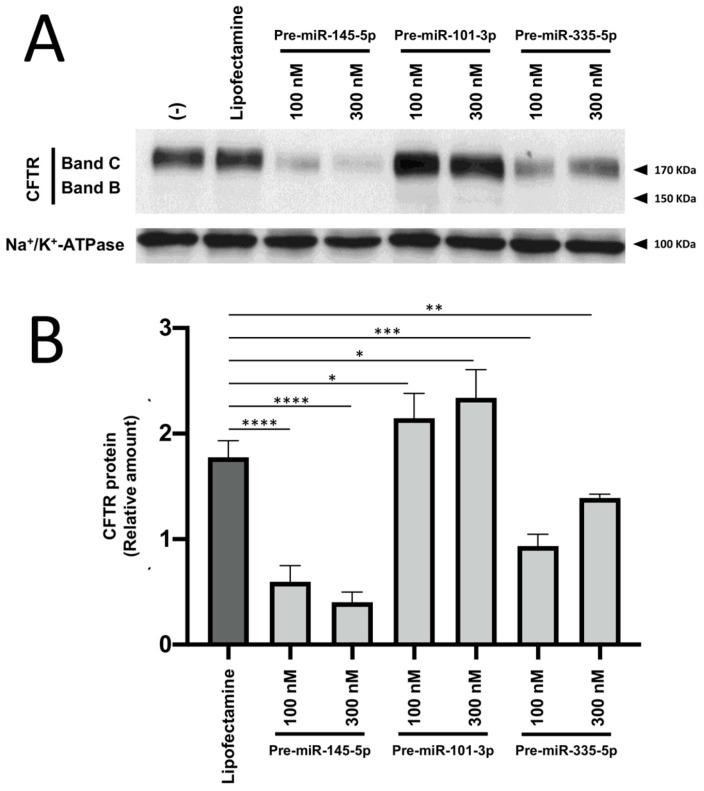
Effects of treatment of Calu-3 cells with pre-miR-145-5p, miR-101-3p, and miR-335-5p. Treatment was 72 h. (**A**). Representative Western blotting gel images. (**B**). Relative values of CFTR content with respect to Na+/K+-ATPase. The uncut version of the gel is shown in Appendix A. (-) = untreated cells. * *p* < 0.05, ** *p* < 0.01, *** *p* < 0.001, and **** *p* < 0.0001.

**Figure 3 ncrna-09-00029-f003:**
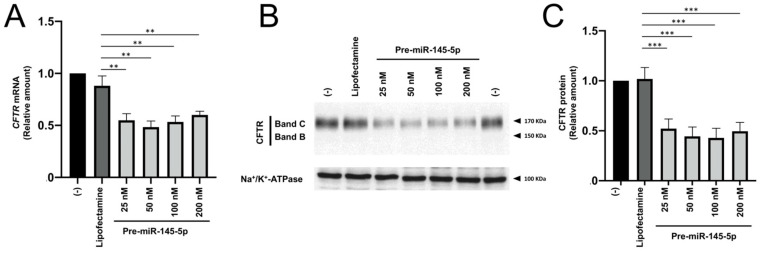
Effects of treatment of Calu-3 cells with increasing concentrations of pre-miR-145-5p. The length of treatment was 72 h. (**A**): RT-qPCR analysis; (**B**): representative Western blotting. (**C**): relative values of CFTR content with respect to Na+/K+-ATPase. The uncut version of the gel is shown in Appendix A. (-) = untreated cells. ** *p* < 0.01, *** *p* < 0.001.

**Figure 4 ncrna-09-00029-f004:**
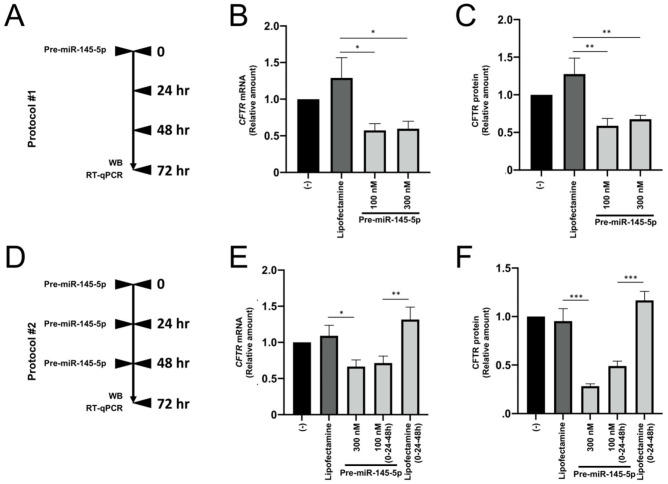
Effects of treatment of Calu-3 cells with pre-miR-145-5p, following protocol #1 (**A**–**C**) and protocol #2 (**D**–**F**). (**A**). Scheme of protocol #1. (**B**). Relative amounts of *CFTR* mRNA (RT-qPCR) in treated Calu-3 cells. (**C**). Relative content of the CFTR protein in treated Calu-3 cells. Treatment was 72 h (see panel **A**). (**D**). Scheme of protocol #2. (**E**). Relative amounts of *CFTR* mRNA (RT-qPCR) in treated Calu-3 cells. (**F**). Relative content of the CFTR protein in treated Calu-3 cells. Treatment was 72 h (see panel **D**). (-) = untreated cells. * *p* < 0.05, ** *p* < 0.01, *** *p* < 0.001.

**Figure 5 ncrna-09-00029-f005:**
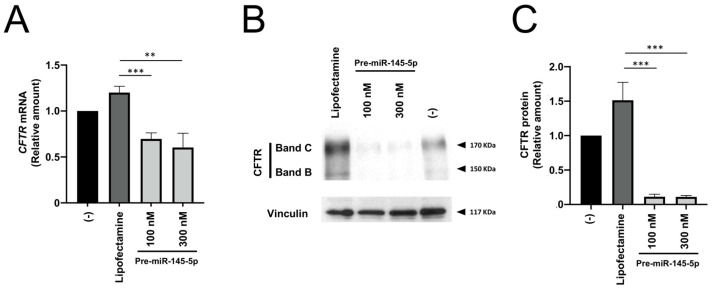
Effects of treatment of human intestinal Caco-2 cells with pre-miR-145-5p following protocol #1. (**A**). Relative amounts of *CFTR* mRNA (RT-qPCR) in pre-miR-145-5p treated Caco-2 cells. Treatment was 72 h. (**B**). Representative Western blotting. (**C**). Relative content of the CFTR protein. The uncut version of the gel is shown in Appendix A. (-) = untreated cells. ** *p* < 0.01, *** *p* < 0.001.

**Table 1 ncrna-09-00029-t001:** The pre-miR miRNAs precursors list employed in Calu-3 and Caco-2 cell treatments.

Pre-miR^TM^ miRNA Precursor	ID
has-pre-miR-145-5p	PM11480
has-pre-miR-101-3p	PM11414
has-pre-miR-335-5p	PM10063

## Data Availability

Most of the data obtained are included in the manuscript and Appendix A. Further information and data will be made freely available by the corresponding authors upon reasonable request.

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
