# Peer review of "The Cystic Fibrosis Transmembrane Conductance Regulator Gene (CFTR) Is under Post-Transcriptional Control of microRNAs: Analysis of the Effects of agomiRNAs Mimicking miR-145-5p, miR-101-3p, and miR-335-5p"

_ncrna, 2023, doi:10.3390/ncrna9020029_

Round 1

Reviewer 1 Report

The main question of this study was to investigate if molecules mimicking the activity of pre-miR-145-5p, pre-miR-335-5p and pre-miR-101-3p influence CFTR expression in bronchial epithelial Calu-3 cells and to discuss if decreasing CFTR expression may have possible translational applications in pre-clinical studies.

The topic is original and focuses on inhibiting CFTR by targeting miRNA using agomiRs. However, if such inhibition could be useful in the clinical setting was not shown. The authors speculated in the Discussion that this may be applied in the treatment of polycystic kidney diseases, cancer or some infectious diseases (Vibrio cholera or Sars-Cov2 infection), but it was not analyzed in their study.

This study showed that pre-miR-145-5p is effective in inhibiting CFTR expression and that the inhibition of pre-miR-145 is higher in comparison with the other studied miRNAs (pre-miR-335-5p and pre-miR-101-3p).

It is not clear why so many different concentration of AgomiR for pre-miR-145 were used (starting from 25 nm up to 300 nM which they stated was the final concentration recommended by the manufacturer) and why the authors selected the highest whereas the lowest concentration (25nM) showed similar efficacy? Could the Authors explain this in the methods section?

Moreover, it is not clear why the authors used two different protocols (single and multiple addition of inhibitor)? It is known that the transfection using inhibitors does not require multiple additions and the manufacturer recommendations include single treatment and they guarantee that the effect of inhibition lasts for 72 hours. So it is not clear why the authors decided to check other experimental protocols if the recommended one is working well? Could you explain this approach?
5. Are the conclusions consistent with the evidence and arguments presented and do they address the main question posed?

The conclusions seem over interpreting the results and stating that “A strength of our study is that the translational applications of our results in the context of“miRNA therapeutics” is very high” is too optimistic and is not based on the results (lines 156-157). Similarly, the anti-cancer potential of that treatment was not analyzed in this study, so it is impossible to state that it can exert an anticancer effect. Moreover, its utility in infectious diseases was also not confirmed as the authors did not analyse the efficacy of inhibition of CFTR in Caco-2 cell line during infection. They showed that CFTR is inhibited by pre-miR-145-5p in this cell line, but it was not shown if this could be effective/sufficient in the treatment of infectious diarrhoea or Sars-Cov2 infection. More cautious statements in the Discussion are recommended that are based on the obtained results.

Author Response

Reviewer #1.

Comments and Suggestions for Authors

The main question of this study was to investigate if molecules mimicking the activity of pre-miR-145-5p, pre-miR-335-5p and pre-miR-101-3p influence CFTR expression in bronchial epithelial Calu-3 cells and to discuss if decreasing CFTR expression may have possible translational applications in pre-clinical studies.
The topic is original and focuses on inhibiting CFTR by targeting miRNA using agomiRs. However, if such inhibition could be useful in the clinical setting was not shown. The authors speculated in the Discussion that this may be applied in the treatment of polycystic kidney diseases, cancer or some infectious diseases (Vibrio cholera or Sars-Cov2 infection), but it was not analyzed in their study.

Answer. We thank the reviewer for her(his) comments and we hope that the changes made will be useful in clarifying the points raised.

Point 1. This study showed that pre-miR-145-5p is effective in inhibiting CFTR expression and that the inhibition of pre-miR-145 is higher in comparison with the other studied miRNAs (pre-miR-335-5p and pre-miR-101-3p).
It is not clear why so many different concentration of AgomiR for pre-miR-145 were used (starting from 25 nm up to 300 nM which they stated was the final concentration recommended by the manufacturer) and why the authors selected the highest whereas the lowest concentration (25nM) showed similar efficacy? Could the Authors explain this in the methods section?

Answer. Preliminary experiments indicated that pre-miR-101-3p exhibited low activity in inhibiting CFTR. Therefore, we have been forced to use 300 nM concentration in order to compare the activity of the three pre-miRNAs (145, 101 and 335) using the same concentration range. We explain this by adding the following sentence: “The treatment with high pre-miR concentrations was performed in order to compare ……requiring comparison at high concentration (i.e. 300 nM) (page 3, lines 76-80). In the materials and methods section, we have clarified that we followed the manufacturer concentrations for lipofectamine (page 7, lines 239-240).

Point 2. Moreover, it is not clear why the authors used two different protocols (single and multiple addition of inhibitor)? It is known that the transfection using inhibitors does not require multiple additions and the manufacturer recommendations include single treatment and they guarantee that the effect of inhibition lasts for 72 hours. So, it is not clear why the authors decided to check other experimental protocols if the recommended one is working well? Could you explain this approach?
Answer. This is a right observation and we agree with the reviewer that the two protocols should be explained. To this aim, the following sentence has been added: “The two protocols were considered taking into account the possibility that ….. therapeutic applications of biomolecules” (page 4, lines 108-111). 

Point 3. Are the conclusions consistent with the evidence and arguments presented and do they address the main question posed? The conclusions seem over interpreting the results and stating that “A strength of our study is that the translational applications of our results in the context of“miRNA therapeutics” is very high” is too optimistic and is not based on the results (lines 156-157). Similarly, the anti-cancer potential of that treatment was not analyzed in this study, so it is impossible to state that it can exert an anticancer effect. Moreover, its utility in infectious diseases was also not confirmed as the authors did not analyse the efficacy of inhibition of CFTR in Caco-2 cell line during infection. They showed that CFTR is inhibited by pre-miR-145-5p in this cell line, but it was not shown if this could be effective/sufficient in the treatment of infectious diarrhoea or Sars-Cov2 infection. More cautious statements in the Discussion are recommended that are based on the obtained results.

Answer. We followed the suggestion of the reviewer and limited over-interpretation of the results. To this aim, the following sentence has been extensively modified: “We suggest that the translational applications of our results to “miRNA therapeutics” should be considered a strength of our study; in this context, the use of pre-miR-145-5p deserves further pre-clinical studies, to be performed using appropriate pathological model systems” (page 6, lines 170-173). As far as the lack of experimental evidences supporting clinically-relevant effects of treatment with the pre-miR-145-5p, we agree that this will be a key focus of future studies. We clarified this by adding the sentence “These results allow proposing the employment of pre-miR-145-5p in pre-clinical studies focusing on pathologies in which CFTR inhibition should be considered a clinically relevant target” (page 6, lines 160-162). However, some pre-clinical information has been already reached. The effects of pre-miR-145 on pre-clinical settings have been already reported. We commented on this by adding the sentence “These published studies, together with our results should stimulate …. CFTR down-regulation, and effects on in vitro tumor phenotype” (page 6, lines 190-193). As far as the effects on infectious diseases, we have already reported a miR-145-5p mediated inhibition of SARS-CoV-2 life cycle. We commented on that by including the sentence “This was formally demonstrated in SARS-CoV-2 infected Calu-3 cells ……. down-regulation of CFTR might be associated with inhibition of SARS-CoV-2 life cycle” (page 7, lines 217-219). We have also discussed the association between the severity of COVID-19 and expression levels of miR-145-5p by including the sentence “Interestingly, using differential expression analysis of microarray data, hsa-miR-145-5p was identified among the top differentially expressed transcripts in severe versus asymptomatic/mild cases. The microarray data confirmed miR-145-5p down-regulation in the severe cases, while miR-145-5p was found up-regulated in the mild cases” (page 7, lines 219-223). The new reference #58 was added to support our discussion on this point.    

Reviewer 2 Report

Comment:1

Line 17: Quantifies, check if this is OK?

Comment:2

Figure-2A, 3B and 5B please indicate the molecular weights. Also mention the complete names of your miRNAs (Pre-miR-145-5p, Pre-miR-101-3p, and Pre-miR-335-5p)

Please follow the same in the Supplementary files as well.

Comment:3

Figure-2B, 3A, 3C, 4B, 4C, 4E, 4F, 5A, and 5C in all these bar graphs the color scheme or the shading of the bars is a bit confusing. Either please follow the same shading or same coloring for the same concentration or simply do not follow the shading of the bar graphs as the concentrations were clearly mentioned below.

For example in Fig:3A and 3C, the coloring or shading of the bars for 50 nm is not the same as in Figures 4B, 4C, 4E, and 4F.

Comment:4 

In westerns, please indicate what the symbol on the first lane indicates.

Comment:5

Figure 3A, why there are 2 controls on western blot indicated in horizontal ( ) which is not reflecting in the quantification of Figure 3C.

Figure 5B the order of samples in western blot is not similar to the quantification in 5C.

Author Response

Reviewer #2.

Comments and Suggestions for Authors

Point 1

Line 17: Quantifies, check if this is OK?

Answer. We changed “quantifies” in “quantified”. Thanks for picking this typing error.

Point 2

Figure-2A, 3B and 5B please indicate the molecular weights. Also mention the complete names of your miRNAs (Pre-miR-145-5p, Pre-miR-101-3p, and Pre-miR-335-5p)

Please follow the same in the Supplementary files as well.

Answer. The molecular weights have been indicated in the alongside the gels images. In addition, the complete names of the pre-miRNAs have been added. We have included these issues also in Supplementary Figures.

Point 3

Figure-2B, 3A, 3C, 4B, 4C, 4E, 4F, 5A, and 5C in all these bar graphs the color scheme or the shading of the bars is a bit confusing. Either please follow the same shading or same coloring for the same concentration or simply do not follow the shading of the bar graphs as the concentrations were clearly mentioned below.

For example in Fig:3A and 3C, the coloring or shading of the bars for 50 nm is not the same as in Figures 4B, 4C, 4E, and 4F.

Answer. We have eliminated the shading in order to minimize the possible confusion. In the revised version only controls untreated cells (-) bars are black, bars relative to cell treated only with lipofectamine are dark gray, while bars relative to all the treated samples are in the same light gray colour. As the reviewer noticed, the concentrations were clearly mentioned below.

Point 4 

In westerns, please indicate what the symbol on the first lane indicates.

Answer. The symbol (-) indicated untreated cells. This was mentioned in the legends of Figures 2, 3, 4 and 5.

Point 5

Figure 3A, why there are 2 controls on western blot indicated in horizontal ( ) which is not reflecting in the quantification of Figure 3C.

Answer. We thank the reviewer for this observation. The point of the control (-) indicated in Figure 3C is the mean of controls untreated samples analyzed in western blotting experiments. In some gels we load the control (-) samples in two or more different lanes surrounding the treated ones.

Figure 5B the order of samples in western blot is not similar to the quantification in 5C.

Answer. To facilitate a better comparison of the results, we have preferred to present the normalized relative content of the CFTR protein (Figure 5C) obtained with western blotting using the same order of the results presented for CFTR mRNA obtained with RT-qPCR (Figure 5A).

Reviewer 3 Report

The manuscript titled “The Cystic Fibrosis Transmembrane Conductance Regulator Gene (CFTR) is under post-transcriptional control of microRNAs: analysis of the effects of agomiRNAs mimicking miR-145-5p, miR-101-3p and miR-335-5p” by Papi et al., explains the production of CFTR mRNA and its proteins and CFTR protein application in several pathological conditions.  

Authors have published several articles recently with similar studies. For example, Papi et al., Int. J. Mol. Sci. 2022; Fabbri et al, Molecules, 2017; Gasperello et al, Data Brief, 2021; Fonotti et al., Am. J. Respir. Crit. Care Med. 2019; Fabbri et al., Eur. J. Med. Chem. 2021; Tamanini et al, Biomedicines 2021.

11. Although the earlier publications by authors show relevant information, authors could explain the novelty of this manuscript.

22. Also, authors could explain how this manuscript is different from the recent published article by authors such as Papi et al., Int. J. Mol. Sci. 2022. This manuscript is quite similar in introduction, methods and mostly results. Please explain.         

Author Response

Reviewer #3.

Comments and Suggestions for Authors

The manuscript titled “The Cystic Fibrosis Transmembrane Conductance Regulator Gene (CFTR) is under post-transcriptional control of microRNAs: analysis of the effects of agomiRNAs mimicking miR-145-5p, miR-101-3p and miR-335-5p” by Papi et al., explains the production of CFTR mRNA and its proteins and CFTR protein application in several pathological conditions. 

Authors have published several articles recently with similar studies. For example, Papi et al., Int. J. Mol. Sci. 2022; Fabbri et al, Molecules, 2017; Gasperello et al, Data Brief, 2021; Fonotti et al., Am. J. Respir. Crit. Care Med. 2019; Fabbri et al., Eur. J. Med. Chem. 2021; Tamanini et al, Biomedicines 2021.

Point 1. Although the earlier publications by authors show relevant information, authors could explain the novelty of this manuscript.

Answer. This is an important point and we are grateful to the reviewer for raising it. To clarify this point the following sentence has been added “In this context, the major focus of previously published studies [20-24, 27] was on hyper-expressing CFTR with the use of antagomiRNA molecules targeting miRNAs down-regulating CFTR (for instance miR-145-5p). This might be of interest for the therapy of Cystic Fibrosis. In those reports, the issue and the translational potential of inhibiting CFTR were not considered. On the contrary ….” (pages 5/6, lines 151-156).

Point 2. Also, authors could explain how this manuscript is different from the recent published article by authors such as Papi et al., Int. J. Mol. Sci. 2022. This manuscript is quite similar in introduction, methods and mostly results. Please explain.

Answer. The manuscript published by Papi et al (IJMS) was based on PNAs targeting CFTR-regulatory miRNAs (the antisense approach); on the contrary, the present paper addresses for the first time (at least in our laboratory) the use of pre-miRNA for miRNA replacement therapy. Of course, the possible translational application in therapeutic protocols will deal with totally different pathologies, and totally different objectives (the objective of the antisense/antimiRNA approach is CFTR upregulation, on the contrary the objective of the miRNA replacement therapy is CFTR inhibition). This was already stated on page 2, lines 52-59, and is now also explained in the new sentences on pages 5/6, lines 151-155.

Round 2

Reviewer 3 Report

The manuscript titled “The Cystic Fibrosis Transmembrane Conductance Regulator Gene (CFTR) is under post-transcriptional control of microRNAs: analysis of the effects of agomiRNAs mimicking miR-145-5p, miR-101-3p and miR-335-5p” by Papi et al., explains the production of CFTR mRNA and its proteins and CFTR protein application in several pathological conditions. Now, authors addressed the questions and changed manuscript accordingly. I recommend accepting this manuscript.